# Visual Function and Ophthalmological Findings in CHARGE Syndrome: Revision of Literature, Definition of a New Clinical Spectrum and Genotype Phenotype Correlation

**DOI:** 10.3390/genes12070972

**Published:** 2021-06-25

**Authors:** Roberta Onesimo, Daniela Ricci, Cristiana Agazzi, Simona Leone, Maria Petrianni, Lorenzo Orazi, Filippo Amore, Annabella Salerni, Chiara Leoni, Daniela Chieffo, Marco Tartaglia, Eugenio Mercuri, Giuseppe Zampino

**Affiliations:** 1Rare Diseases Unit, Fondazione Policlinico Universitario Gemelli-IRCCS, 00168 Rome, Italy; cristiana.agazzi01@icatt.it (C.A.); chiara.leoni@policlinicogemelli.it (C.L.); giuseppe.zampino@unicatt.it (G.Z.); 2Pediatric Unit, Fondazione Policlinico Universitario Agostino Gemelli-IRCCS, 00168 Rome, Italy; 3National Centre of Services and Research for the Prevention of Blindness and Rehabilitation of Low Vision Patients-IAPB Italia Onlus, 00185 Rome, Italy; daniela.ricci@guest.policlinicogemelli.it (D.R.); simona.leone@guest.policlinicogemelli.it (S.L.); maria.petrianni@guest.policlinicogemelli.it (M.P.); orazi.lorenzo@gmail.com (L.O.); filippo.amore@policlinicogemelli.it (F.A.); 4Pediatric Neurology Unit, Fondazione Policlinico Agostino Gemelli-IRCCS, 00168 Rome, Italy; EugenioMaria.Mercuri@unicatt.it; 5Ophthalmology Unit, Fondazione Policlinico Universitario Agostino Gemelli-IRCCS, 00168 Rome, Italy; annabella.salerni@policlinicogemelli.it; 6Clinical psychology Unit, Fondazione Policlinico Universitario Agostino Gemelli-IRCCS, 00168 Rome, Italy; danielapiarosaria.chieffo@policlinicogemelli.it; 7Genetics and Rare Diseases Research Division, Ospedale Pediatrico Bambino Gesù IRCCS, 00168 Rome, Italy; marco.tartaglia@opbg.net; 8Pediatric Neurology Unit, Catholic University of Sacred Heart, 00168 Rome, Italy

**Keywords:** CHARGE syndrome, coloboma, visual function, *CHD7*, genotype–ocular-phenotype, VISIOCHARGE, rare diseases

## Abstract

CHARGE syndrome (CS) is a rare genetic disease causing multiple anatomical defects and sensory impairment. Visual function is usually reported by caregivers and has never been described with a structured behavioral assessment. Our primary objective was to describe ocular abnormalities, visual function and genotype–ocular-phenotype correlation in CS. A prospective monocentric cohort study was performed on 14 children with CS carrying pathogenic *CHD7* variants. All children underwent ophthalmological evaluation and structured behavioral assessment of visual function. The VISIOCHARGE questionnaire was administered to parents. Colobomas were present in 93% of patients. Genotype–phenotype correlation documented mitigated features in a subset of patients with intronic pathogenic variants predicted to affect transcript processing, and severe features in patients with frameshift/nonsense variants predicting protein truncation at the N-terminus. Abnormal visual function was present in all subjects, with different degrees of impairment. A significant correlation was found between visual function and age at assessment (*p*-value = 0.025). The present data are the first to characterize visual function in CS patients. They suggest that hypomorphic variants might be associated with milder features, and that visual function appears to be related to age. While studies with larger cohorts are required for confirmation, our data indicate that experience appears to influence everyday use of visual function more than ocular abnormalities do.

## 1. Introduction

CHARGE syndrome (CS; OMIM 214800) is a rare genetic disorder with an estimated incidence of 1/12,000. The term ‘CHARGE’ is an acronym for the most striking clinical features of the syndrome, involving coloboma, heart defects, choanal atresia, retarded growth and development, genitourinary malformation and ear abnormalities [1].

CS diagnosis is based on major and minor clinical criteria [2]. Molecular testing in clinically diagnosed individuals identifies variants in *CHD7* (OMIM 608892) in 70–90% of cases [3]. 

The broad range of organs and systems affected makes management of CS very challenging, with clinicians from multiple disciplines involved in clinical surveillance [4,5]. Children with CS suffer from multiple sensory disabilities. The auditory system is invariably affected, with ear abnormalities and sensorineural hearing loss [6,7]. The visual system is also frequently affected, with coloboma and other ocular abnormalities being present in up to 90% of patients [8]. Coloboma is typically chorioretinal, but it can also involve the eyelid, iris and optic disc, and it is usually bilateral. The involvement of the macula is responsible for significant reductions in central visual acuity. Anterior segment abnormalities such as microcornea and cataracts are also commonly described and may be associated with other less frequent features, including severe refractive errors, strabismus and ptosis [9]. Retinal detachment rarely occurs and represents a complication worsening the visual impairment. Visual acuity is reported as lower than 20/60 in evaluable subjects [8,9,10,11,12], but more detailed information on acuity and other aspects of visual function is limited because of the difficulty in obtaining the full participation of CS patients who, by definition, have a complex clinical presentation. 

Recently, Martin et al. proposed a self-administered questionnaire entitled VISIOCHARGE and found relatively good visual skills in everyday life in patients with CS [13], but the extent of impairment of visual function cannot be easily appreciated by considering reported measures only.

In the last decade, the visual competences of subjects with other syndromic conditions have been described using a specific protocol of behavioral assessment adapted for patients with multiple disabilities [14,15]. Based on these considerations, the main aim of our study was to use a more systematic assessment of ocular abnormalities and visual function with both a structured behavioral evaluation and a parent/carer-reported questionnaire (VISIOCHARGE) in a cohort of CS patients carrying *CHD7* pathogenic variants. We also aimed to establish relationships between ocular abnormalities and visual function and to explore genotype–ocular-phenotype correlations.

## 2. Materials and Methods

### 2.1. Study Population

Fourteen individuals (8 males, 6 females; age range 2.6–26 years, mean: 10.9 ± 7.6) with CS, regularly followed in the Rare Disease Unit of the Pediatrics Department, Fondazione Policlinico Agostino Gemelli-IRCCS, Rome, Italy, were consecutively enrolled from February 2019 to March 2020. 

All patients underwent a multidisciplinary assessment including an examination with a pediatrician with expertise in disability and an experienced clinical geneticist. All subjects met the clinical criteria for CS and had been molecularly confirmed to carry heterozygous *CHD7* variants classified as pathogenic/likely pathogenic. Patients’ demographic, clinical and genetic data are shown in Table 1. 

All patients were referred to the National Centre of Services and Research for the Prevention of Blindness and Rehabilitation of Low Vision Patients for a detailed ophthalmological examination and a behavioral assessment of visual function. Informed consent was obtained from all guardians and data were collected anonymously. The study was approved by the local Research Ethical Committee.

### 2.2. Methods 

All patients underwent an ophthalmological evaluation and a structured behavioral assessment of visual function. The VISIOCHARGE questionnaire was administered to the caregivers of all CS patients.

### 2.3. Ophthalmologic Evaluation 

The ophthalmological examination consisted of slit lamp examination and cycloplegic refraction fundoscopy in all patients. Retinography and optical coherence tomography (OCT) were performed in actively participating patients. 

The findings were classified according to the morphology. The ophthalmological grading of the severity was based on the retinal extension of the colobomas and the involvement of the macula or the optic disc. The severity of retinal and optical nerve abnormalities was scored (0–4) according to increasing severity:
0 = no retinal abnormalities (N); 1 = mild (optic nerve pallor (ONP) or small optic nerve coloboma (SONC)); 2 = moderate (small chorioretinal coloboma (SCCR));3 = severe (large chorioretinal coloboma (LCCR) or large optic nerve coloboma (LONC)); 4 = extremely severe (chorioretinal coloboma with macular involvement (CCRM)/giant coloboma (GC)/microphthalmos (Microph)/retinal detachment (RD)). 

Small optic nerve coloboma: coloboma involving less than 1/3 of the optic disc surface.

Optic nerve coloboma: coloboma involving more than 1/3 of the optic disc surface. 

Small retinal coloboma: coloboma size in a maximum range of 2 optic disc diameters.

Moderate coloboma: coloboma size ranging from 2 optic disc diameters to maximum of 1/4 of the whole retinal surface. 

Large coloboma: coloboma size of more than ¼ of the whole retinal surface.

A patient’s total score, ranging from 0 to 8, was calculated by adding the results of both eyes. A score between 0 and 5 was defined as mild, and a score between 6 and 8 was defined as severe.

### 2.4. Visual Function Behavioral Assessment 

The visual function behavioral assessment consisted in the assessment of various aspects:

Ability to fix (stable, unstable, absent) [16], to track horizontally, vertically and in a full circle (complete, head compensation, absent) [16] and perform horizontal and vertical saccades (normal, head compensation, absent). These were elicited using a black/white, colored or lit target.

Visual acuity was assessed using the Teller acuity card procedure [17,18,19], LEA Symbols optotypes or Early Treatment Diabetic Retinopathy Study (ETDRS) charts [20,21] according to the subject’s age and level of active participation.

Attention at distance was assessed using a high-contrast target. It was considered normal at 3 m and abnormal at < 3 m in subjects older than 3 months of age. 

Visual fields were assessed using kinetic perimetry according to the technique described in detail by van Hof-van Duin [22,23] and were defined as normal, reduced symmetrically, reduced asymmetrically or not evaluable. 

Contrast sensitivity was assessed using the Hiding Heidi contrast sensitivity test or the Pelli Robson board according to the subject’s age and active participation and was defined as normal, reduced or not evaluable.

Stereopsis was assessed using the Frisby stereotest or the Lang stereotest according to the subject’s active participation and was defined as normal, abnormal or absent. 

Strabismus was evaluated with the Hirschberg test, Krimsky test or prismatic cover test according to the subject’s active participation.

A visual function total score, with higher scores indicating worse visual function, was used in order to compare functional vision to ocular abnormalities. The scores could range from 35 (all aspects involved) to 0 (all items passed).

### 2.5. VISIOCHARGE Questionnaire

The VISIOCHARGE questionnaire [13] was validated in the Italian language [24] and administered to the parents of all patients. It includes 30 items, divided in three categories: (1) parental evaluation of global vision, designed to assess parents’ opinion about the importance of their child’s visual impairment and its effect on everyday life; (2) evaluation of distance vision; and (3) evaluation of near vision. Additional questions included information about the last ophthalmological visit, educational level and age of walking acquisition. Scores were expressed as decimals between 0 and 1 and calculated according to Martin et al. [13].

### 2.6. Statistical Analysis

Descriptive statistics, such as mean and standard deviation, were used to characterize the cohort. In order to correlate anatomic severity with functional data, simple linear regression was used. Patients were divided in two groups according to the ocular abnormalities severity and age and then compared using the Mann–Whitney U test. A two-sided *p*-value of less than 0.05 was considered significant.

## 3. Results

Details of the ophthalmological and behavioral assessment and genetic findings are reported in Table 2.

### 3.1. Ophthalmological Evaluation

All patients had some degree of ocular abnormality, as summarized in Table 1 and Table 2. Refractive disorders could be evaluated in 13 patients (24 eyes, excluding two eyes with microphthalmos). Myopia was documented in eight subjects (15 eyes), five eyes showed a defect greater than 3 diopters and hypermetropia was found in nine eyes from five subjects. At the examination, eight patients wore optical correction as previously prescribed. The four patients with bilateral CCRM also underwent OCT, which showed absence of physiological foveal depression at the posterior pole. In all patients but three, the abnormalities were bilateral. 

The total scores reporting the overall severity of ocular abnormalities ranged between 2 and 8.

### 3.2. Visual Function Behavioral Assessment 

All patients underwent the assessment of visual function. Visual field was not evaluable in two patients (14%), and stereopsis was not evaluable in one patient (7%). All patients had some degree of impairment in more than one aspect of visual function. 

Ability to fix was present in all patients, stable in 9/14 patients and unstable in the remaining five subjects. Seven patients reacted to the black/white or colored stimulus, while the other seven needed a lit target. Ability to track was variable with increasing difficulties from horizontal to circle. 

Tracking for a circle needed head compensation in 5/14 patients and was absent in the remaining 9. 

Horizontal saccades were present in 2/14 patients, needed head compensation in 8 and were absent in the remaining 4. 

Vertical saccades needed head compensation in nine patients and were absent in the remaining five. 

Visual acuity was normal in 2/14 patients, mild low vision was found in 10 and severe low vision in the remaining 2. 

Attention at distance was normal (3 m) in 7/14 patients and was reduced in the others. 

Visual fields were normal and symmetrical in 1/14 patients, reduced in 10 (4 symmetrical and 6 asymmetrical) and not evaluable in the remaining 3 subjects. 

Contrast sensitivity was normal in 6/14 patients and reduced in the remaining 8 subjects. 

Stereopsis was present and adequate in 1/14 patients, absent in 12 patients and not evaluable in the remaining subject. 

All patients presented some degree of alterations in ocular motility. Eleven of the fourteen patients presented strabismus. Horizontal nystagmus was present in 5/14 patients and ocular dyspraxia also in 5. 

The visual function total score, with higher scores indicating worse visual function, ranged between 3 and 25 (mean 12.64, SD 5.77). Details are reported in Table 2. 

### 3.3. VISIOCHARGE Questionnaire

On the items assessing global vision, designed to assess parents’ opinion about the importance of their child’s visual impairment and its effect on everyday life, the parents of 11/14 patients (78%) reported difficulties related to vision, with 4/11 (28%) severely bothered by it.

On the items evaluating distance vision and in those evaluating near vision, the mean scores of distance vision and near vision were 0.49 ± 0.24 and 0.6 ± 0.17, respectively. 

The details are reported in Table A1 (see Appendix A).

### 3.4. Visual Function Behaviural Assessment and Age at Assessment

Patients younger than 7 years had a more severe visual impairment (mean score = 16.6) than those older than 7 (mean score = 10.4) (*p*-value= 0.025) (Figure 1a).

### 3.5. Ocular Abnormalities and Visual Function Behavioral Assessment 

All patients had some ocular abnormalities and some functional impairment, but there was no association between severity of ocular abnormalities and severity of visual functional impairment (*p*-value = 0.31, R^2^ = 0.08) (Figure 1b).

### 3.6. Ocular Abnormalities and VISIOCHARGE

There was no correlation between severity of ocular abnormalities and global VISIOCHARGE data in terms of visual acuity, distance vision, near vision scores and overall ability score (*p*-value > 0.1). 

### 3.7. Genotype–Ocular-Phenotype Correlation

*CHD7* variants data were available for all patients. The pathogenic variants included nonsense (*n* = 5) and frameshift (*n* = 4) variants predicting anticipated protein truncation (*n* = 9), splice site changes affecting transcript processing (*n* = 4) and one missense variant (Table 1 and Table 2). Variants were spotted throughout the coding sequence of the gene and their flanking intronic regions.

Among the nine patients carrying truncating variants, due to either frameshift or nonsense changes, CCRM was found in four patients bilaterally, with all showing pathogenic variants affecting the N-terminus of the protein, and in one patient unilaterally with a nonsense variant in exon 29. In this subgroup, single patients showed LCCR (RE)/RD (LE), LONC bilaterally and SONC occurring either bilaterally or unilaterally (subject with unilateral CCRM). Finally, one patient with a nonsense variant affecting the C-terminus showed milder ocular defects, having normal eye morphology (N) on the right and SCCR on the left. The individual with a missense variant presented SCCR (RE) and CCR (LE). Finally, the ocular severity in the four subjects with variants affecting splice sites was heterogenous, with bilateral CCRM observed in a patient with a variant affecting a +2 position and milder ocular phenotypes in the remining patients with variants affecting positions +5/−4, including one presenting ONP bilaterally, one with SONC (RE) and N eye (LE) and the remaining patient who was affected by a GC involving the LE and completely normal eye morphogenesis (N) of the RE.

## 4. Discussion

Ocular abnormalities, and large chorioretinal colobomas in particular, are a major feature of CS, although it is known that the eye phenotype does not have a full penetrance. Our study confirmed in a genetically characterized cohort of patients with CS that ocular abnormalities are a constant feature of this disorder, with 93% of the affected subjects having colobomas. Additionally, our study showed that ocular motility abnormalities, including strabismus, nystagmus and oculomotor dyspraxia, are frequent findings in CS. These data are in agreement with previous studies that, contrarily, included patients in whom clinical diagnosis was not always molecularly confirmed (Table 3).

All patients in this study carried *CHD7* variants classified as pathogenic. As expected, premature truncation of the protein was the most frequently predicted effect of variants, followed by splice site mutations and missense mutations more rarely [26,27,28]. Even though no structured genotype–ocular-phenotype correlation could be noted due to the small sample size, we observed a different clustering in terms of severity of ocular abnormalities according to the type of mutation. Truncating mutations, either frameshift or nonsense, affecting the N-terminus of the protein were associated with severe ocular findings, such as large retinal colobomas involving choroid, macula and/or the optic nerve; the closer the pathogenic variant was from the N-terminus +6+69−\6, the worse was the anatomic defect. On the other hand, splice site variants affecting positions different from ±1/±2 were associated with a milder ocular phenotype, which could be explained by a milder impact of these splice site variants on transcript processing.

Despite the fact that ocular abnormalities have been reported to be associated with impairment of visual function [10], thus far, this has not been systematically explored. One of the aims of our study was to use an integrated approach to establish the impact of ocular abnormalities on visual function in CS patients. Difficulties in performing a complete behavioral clinical assessment depend partly on the level of active participation of CS patients, which is not always sufficient. Most information on visual function is available from ophthalmological assessments or from parent-reported measures [8,9,10,11,13]. 

To our knowledge, this is the first study using a systematic integrated approach consisting of a structured behavioral assessment of visual function in combination with a detailed ophthalmological evaluation in CS patients. The selection of tools for the assessment of different aspects of visual function was made according to age, level of active participation and possibility to be performed even in young children with multisensorial/cognitive impairment. Using this adapted protocol, which is part of our clinical practice for patients with multisensorial disability [14,16], almost all patients were able to complete the structured assessment. By using this strategy, we were able to demonstrate that all patients had at least one aspect of visual function impaired and that 93% had impairments of several aspects. It is of interest that most patients showed normal results only on the less complex aspects of visual function. Although some patients needed more contrasted or well-lit targets, all were able to fix and track horizontally, but upon increasing the complexity of the action, such as tracking vertically and for a circle, many of them had more difficulties. Looking at more complex aspects of visual function, we observed different degrees of impairment. Absent stereopsis and visual field defects were found in almost all patients. Severe impairment of visual acuity was found in 57% of our patients, in agreement with recent studies [7,8]. Contrast sensitivity was reduced in almost half of the patients. As these competences involve the exploration of objects, faces, images and space, it is likely that the impairment of these abilities may influence visuo-cognitive function and social life. 

Interestingly, despite the fact that all patients had some degree of both ocular and functional impairment, when comparing the severity of retinal/optical nerve involvement to the severity of functional abnormalities, no significant correlation was found. The only patient with very mild signs of functional impairment was one of the three with unilateral ocular involvement. In the other two patients with a monocular anatomic defect and in all the others with binocular involvement, some aspects of visual function, such as stereopsis and visual fields, were always affected, irrespective of the severity of ocular abnormalities. Other functional aspects showed more variable results and were also not consistently associated with severity of ocular involvement. The lack of correlation may be partially explained by our relatively small number of patients or our scoring systems, as we know that a severe unilateral defect has a different effect on visual functionality compared to a mild bilateral defect. However, analyzing singular results, we found that in some subjects with severe ocular abnormalities (i.e., score of four bilaterally), the functional results were better than expected. This was more often found in older patients and could be due to a possible improvement over time in relation to environmental influence, with specific training or experience. Of course, we are aware that combining the results of both eyes is a limitation of this study. 

Similar results were also seen when analyzing parents’ responses to the VISIOCHARGE questionnaire [13]. Our VISIOCHARGE findings are comparable to those previously reported (details in Table A1), thus confirming the feasibility of this tool in CS patients. On the items assessing global vision and the effect of the child’s visual impairment on everyday life, 11/14 (78%) reported difficulties related to vision, defined as severe in 4 of the 11 (28%). Parental evaluation of global vision was better in children with milder ocular abnormalities than in those with more severe impairment, but the difference did not reach significance. It is of interest that most patients were reported to have more adequate near vision than distance vision that was frequently described as impaired, and that, similarly to the pilot study, visual acuity was correlated to distance vision and near vision scores. Although these patients presented reduced visual ability, they were still able to perform complex activities, such as using electronic devices including smartphones or digital tablets or reading or seeing written characters of an adequate size.

Our data provide, for the first time, an accurate description of different aspects of visual function in CS patients in relation to ocular abnormalities, parental perception and to genotype. 

In general, we found that despite all patients having some degree of abnormality in all three assessment methods used (ophthalmological, functional behavior and parent/carer-reported measures), none had such a severe functional outcome to prevent subjects from using some aspects of vision in everyday life. Each of the three approaches contributed to define the spectrum and severity of visual impairment that could not always be predicted by the severity of ocular abnormalities. The VISIOCHARGE questionnaire proved to be a very useful tool to initially assess visual abilities in CS patients, while the behavior-adapted protocol provided more detailed information on the functional aspects of vision. These results suggest that behavioral assessment of visual function is feasible and should be performed in CS patients, as it can provide evidence of the impairment of specific aspects of visual function that can be useful when planning rehabilitation programs. 

The possible role of intervention deserves more attention, as in our cohort, older and more experienced subjects who had received training and regular therapy appeared to have better functional results. 

Studies with larger cohorts and possible randomization of rehabilitative intervention are required to confirm the extent to which early recognition and management of sensory deficits may affect long-term global functioning in patients affected by CS. 

## 5. Conclusions

Our study was the first to use a systematic integrated approach to provide detailed description of visual function in CS patients and to associate it to ocular abnormalities, parental perception and to genotype. All patients carried *CHD7* variants classified as pathogenic. A different clustering in terms of severity of ocular abnormalities according to the type of mutation was observed.

## Figures and Tables

**Figure 1 genes-12-00972-f001:**
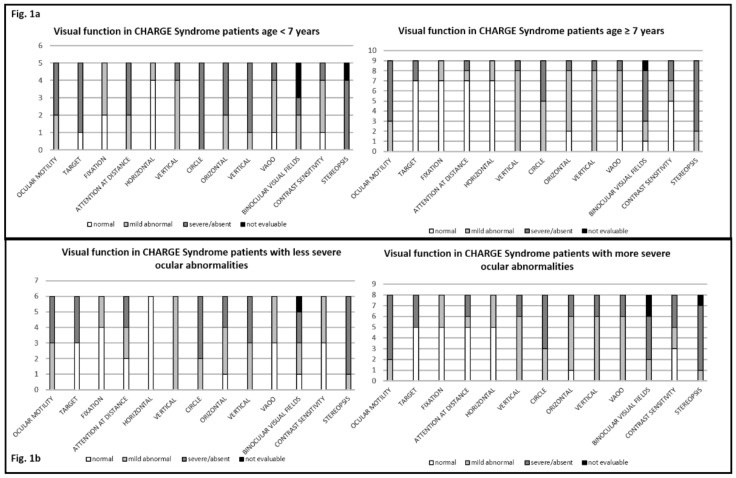
Association between visual impairment and age at assessment (**a**), and ocular abnormalities (**b**) (less severe: score ≤ 4; more severe > 4).

**Table 1 genes-12-00972-t001:** Details of the CS cohort.

Patients (Number) 14		
**Demographics**		
Age (years)	10.98 ± 7.6	
Gender (M)	8	
**Genetics**		
*CHD7* mutation	14	100
Frameshift	4	28.6
Nonsense	5	35.7
Missense	1	7
Splicing	4	28.6
**Clinical features**	**n**	**Percentage (%)**
**Major characteristics**		
Cranial nerve dysfunction	7	50
Choanal anomaly	3	21.4
Hearing loss	14	100
Sensorineural	12	85.7
Conductive	2	14.3
Hearing severity		
Normal	0	0
Mild/moderate hearing loss	3	21.4
Severe to total hearing loss	11	78.6
**Ocular abnormalities**	14	100
Posterior coloboma		
Unilateral	4	28
Bilateral	9	65
Iris coloboma		
Unilateral	2	14
Bilateral	4	28
Optic nerve pallor		
Unilateral	0	0
Bilateral	1	7
Microphthalmos		
Unilateral	2	14
Bilateral	0	0
Retinal detachment		
Unilateral	1	7
Bilateral	0	0
Retinal dystrophy		
Unilateral	1	7
Bilateral	0	0
**Minor characteristics**		
Cardiovascular malformation	8	57.1
Genital hypoplasia	4	28.6
Orofacial cleft	2	14.3
Tracheoesophageal fistula	4	28.6
Developmental delay	14	100
Growth deficiency	3	21.4
Distinctive CHARGE facies	14	100

**Table 2 genes-12-00972-t002:** Details on behavioral visual function assessment and ocular abnormalities.

Pt. ID	1	2	3	4	5	6	7	8	9	10	11	12	13	14
**Age (years)**	2	4	5	5	6	7	7	8	8	12	13	19	26	26
**Stereopsis**	Absent	Absent	Absent	Absent	NE	Absent	Absent	Absent	Normal	Absent	Absent	Absent	Absent	Absent
**Contrast sensitivity**	Normal	Reduced	Reduced	Reduced	Reduced	Reduced	Reduced	Normal	Normal	Reduced	Normal	Normal	Reduced	Normal
**Binocular visual** **field**	60° bilateral	50° bilateral	50° right 20° left	NE	NE	40° right 60° left	10° right 60° left	30° bilateral	90° bilateral	50° right 60° left	10° right 50° left	50° bilateral	NE	60° right 5° left
**Vaoo Logmatr**	0.3	0.7	0.7	1	1.6	1.3	0.55	0.1	0.1	0.7	0.55	0.55	1	0
**Vertical saccades**	Absent	HC	Absent	Absent	Absent	HC	HC	HC	HC	HC	HC	HC	Absent	HC
**Horizontal saccades**	Absent	HC	Absent	HC	Absent	Normal	HC	HC	Normal	HC	HC	HC	Absent	HC
**Circle**	Absent	Absent	Absent	Absent	Absent	Absent	Absent	HC	HC	Absent	HC	HC	Absent	HC
**Vertical tracking**	HC	HC	HC	HC	Absent	HC	HC	HC	HC	HC	HC	HC	Absent	HC
**Horizontal tracking**	Complete	Complete	Complete	Complete	HC	Complete	Complete	HC	Complete	Complete	Complete	Complete	HC	Complete
**Attention at distance**	1 m	2 m	2.5 m	1 m	50 cm	1.5 m	3 m	3 m	3 m	3 m	3 m	3 m	50c m	3 m
**Fixation**	Unstable	Stable	Stable	Unstable	Unstable	Stable	Unstable	Stable	Stable	Stable	Stable	Stable	Unstable	Stable
**Ophth. Staging LE**	1	0	4	1	4	3	4	4	2	4	3	3	4	4
**Ophth. Staging RE**	1	1	1	1	3	3	4	4	0	4	4	2	4	0
**LE fundus**	SONC	N	CCRM + Microph.	ONP	RD	LONC	CCRM	CCRM	SCCR	CCRM	LCCR	CCR	CCRM	GC
**RE fundus**	SONC	SONC	SONC	ONP	LCCR	LONC	CCRM	CCRM	N	CCRM	CCRM + Microph.	SCCR	CCRM	N
***CHD7* nucleotide change predicted amino acid change**	c.4795C>T p.Gln1599Ter	c.2957+5G>A (spl) ^a^	c.5782C>T p.Gln1928Ter	c.2442+5G>A (spl) ^b^	c.5722_5723delAC p.Thr1908ProfsTer17	c.3004C>T p.n1001Ter	c.969-975delAACAA p.Val323TyrfsTer11	c.2509_2512delCATT p.His837ValfsTer5	c.7803C>G p.Tyr2601Ter	c.1163C>G p.Ser230Ter	c.6936+2T>A (spl) ^c^	c.3156T>A p.Ser1052Arg	c.1774delC p.Gln592SerfsTer16	c.7165-4A>G (spl) ^d^

RE, right eye; LE, left eye; Opth., ophthalmologic; HC, head compensation; NE, not evaluable; Microph., microphthalmos; ^a^ Highest SpliceAI Delta score = 0.76 (donor loss). ^b^ Highest SpliceAI Delta score = 0.33 (donor loss). ^c^ Highest SpliceAI Delta score = 0.99 (donor loss). ^d^ Highest SpliceAI Delta score = 0.9 (acceptor gain). (https://spliceailookup.broadinstitute.org/ (accessed on 5 March 2021)).

**Table 3 genes-12-00972-t003:** Ocular abnormalities in CHARGE syndrome: review of the literature and current study.

Review of Anatomical Defects in CHARGE Syndrome								
Ocular Defect	1990	1998	2006	2006	2008	2012	2020	2020
	Russel-Eggitt (*n* = 50)	Tellier (*n* = 47)	Aramaki (*n* = 17)	Jongmans (*n* = 47)	McMain (*n* = 9)	Nishina (*n* = 19)	Martin (*n* = 83)	Current Study (*n* = 14)
**Coloboma**	86%	79%	88%	70%	89%	95%	83%	93%
**Unilateral**	16%	57%	N/S	2%	11%	5%	17%	29%
**Bilateral**	64%	21%	N/S	68%	78%	89%	66%	64%
**Retinochoroidal**	80%	43%	58%	N/S	89%	95%	N/S	71%
**Optic disk**	74%	17%	30%	N/S	N/S	95%	N/S	36%
**Macula**	N/S	N/S	N/S	N/S	N/S	68%	N/S	43%
**Iris**	26%	6%	12%	19%	11%	89%	14%	36%
**Eyelid**	2%	N/S	N/S	N/S	0	0	N/S	0
**Microphthalmos**	42%	34%	N/S	21%	N/S	26%	34%	14%
**Unilateral**	26%	N/S	N/S	N/S	11%	10%	30%	14%
**Bilateral**	16%	N/S	N/S	N/S	N/S	16%	3%	0
**Microcornea**	N/S	N/S	N/S	N/S	11%	21%	N/S	0
**Cataract**	2%	N/S	N/S	N/S	11%	5%	5%	7%
**Retinal detachment**	2%	N/S	N/S	N/S	0	0	7%	7%

N/S, not specified. References: (Aramaki et al., 2006 [25]; Jongmans et al., 2006 [26]; Martin et al., 2020 [13]; McMain et al., 2008 [9]; Nishina et al., 2012 [10]; Russell-Eggitt et al., 1990 [8]; Tellier et al., 1998 [12]).

## Data Availability

The data presented in this study are available on request from the corresponding author. The data are not publicly available due to privacy concerns.

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
