# Peer review of "Visual Function and Ophthalmological Findings in CHARGE Syndrome: Revision of Literature, Definition of a New Clinical Spectrum and Genotype Phenotype Correlation"

_genes, 2021, doi:10.3390/genes12070972_

Round 1

Reviewer 1 Report

The goal of this study by Onesimo et al. was to verify whether CHD7 mutations correlated with ocular malformations and/or visual function in the context of CHARGE syndrome. The authors also addressed the previously unexplored question of whether malformation severity might correlate with degree of visual impairment. They found no evidence of such correlation, but found a trend between malformation severity and type of CHD7 mutation. A statistically significant correlation (although questionable, see below) was only found between visual function and age at assessment, which was interpreted as indicative of adaptation with experience. Although definitive conclusions are limited, mainly because of the small sample size, this study would be of interest for the field provided that the following concerns are properly addressed (in order of appearance):

  • Line 25: “molecularly confirmed CS” should be replaced by “CHD7 mutation-positive CS”, to account for the facts that 1) a relatively important subset of CS cases are negative for CHD7 mutation, and 2) not all individuals with a CHD7 mutation develop CS.
  • Line 46: “in 90%” should be replaced by “in 70-90%”, to account for the variable percentage observed in different cohorts.
  • Line 56: microphtalmia is not restricted to the anterior segment
  • Line 61: “…poor level of collaboration of CS patients…” sounds pejorative. Consider changing it for something like “difficulty in obtaining full participation of CS patients who, by definition, have a complex clinical presentation.”
  • Line 72: 14 patients is not a “relatively large cohort”. And, here also, “molecularly confirmed CS” should be replaced by “CHD7 mutation-confirmed CS”.
  • Line 77: “children” should be replaced by “individuals” to account for the presence of adolescents and adults in the cohort.
  • Lines 83-84: “pathogenic heterozygous CHD7 variants” should be replaced by “heterozygous CHD7 variants classified as pathogenic” to account for the fact that pathogenicity has not been experimentally confirmed.
  • Line 99: consider changing “collaborating” for “actively participating”
  • Lines 119-120: combining the results of both eyes prevent distinguishing between severe unilateral from mild bilateral defects. This should be noted as a limitation.
  • Lines 129, 136, 139, 141 : consider changing “collaboration” for “active participation”
  • Lines 135 to 141: please define the criteria that were used to select specific tests as a function of age and/or degree of participation.
  • Table 2 is very dense, with many instances of words that are cut. Moving the information from the far right column in the footnotes would help gaining space.
  • Line 208: please specify how distance vision and near vision scores are calculated, and indicate what are the worst and best possible scores (0-1?).
  • Section 3.4 and Fig1a (which is too small and hard to read): developmental delay might be a confounding factor here. Also, the rationale for choosing 6 years instead of 8 years as cut-off should be explained. It would make much more sense to split the group in children (2-8y) vs adolescent/young adults (12y+).
  • Line 251: not having a CHD7 mutation does not preclude having a diagnosis for CHARGE
  • Table 3: please avoid cutting author names in the third line.
  • Line 254: “pathogenic CHD7 mutations” should be replaced by “CHD7 mutations classified as pathogenic”.
  • Lines 261-262: “the earlier the mutation occurred” would read better as “the closest the mutation was from the N-terminus”.
  • Line 270: “collaboration of” should be replaced by “active participation from”.
  • Line 276: “collaboration” should be replaced by “active participation”.
  • Lines 300-301: “by the relative small number of our patients partly by the scoring systems.” Would read better as “by our relative small number of patients and/or our scoring systems.”
  • Line 307: Please elaborate about the similarities of the VISIOCHARGE findings between this study and the previously published one. Can the same correlations between visual acuity and distance/near vision scores be seen?
  • Supplementary Table looks incomplete, with many empty boxes

Author Response

Thank you for revising our paper, we have carefully considered all issues and suggestions raised by you.

We provide a point-by-point response to your comments.

Reviewer 2 Report

Onesimo and colleagues nicely describe anatomical and functional ophthalmogical features in a group of 14 individuals with molecularly proven CHARGE syndrome. This is a nice study that adds to the knowledge on the complex CHARGE phenotype.

I have some minor and major comments that are listed below, approximately in the order of the text.

  1. First of all, I wonder whether there might have been selection bias in this study. None of the patients had bilateral anatomically normal eyes. From our experience we know that the eye phenotype does not have a full penetrance. The authors should comment on this in their discussion.
  2. Methods: in order to classify morpholigocal severity the authors add up the scores of both eyes. However a score of 0 + 4 will have a different effect on visual functionality than a score of 2 + 2.
  3. Please refer to table 2 already in the beginning of section 3
  4. Table 2: the order of the features should be in the order of the text. Start with patient Id and age, followed by anatomical features and then functional features.
  5. CHD7, whenever the gene is meant, should be in italics
  6. Table 2: there errors in the row on CHD7 variant. First of all CHD7 can be removed form all cells, since it already is in the title cell. For Id 1 there is a typo, for Id 14 the c. notation is deuplicated. Below the table (lines 200-201) there is an additional variant mentioned. Where does that belong?
  7. Table 2: if splice site prediction programs predict a splice site effect add (spl) after the c. notation. If the splice site effect has been proven, e.g. by a mine-gene essay, add spl without brackets.
  8. Figure 1: explain in the legend what the definition is of “less severe ocular abnormalities” and “most severe ocular abnormalities”.
  9. Paragraph 3.7: I count 5 nonsense, 5 intronic and 4 frameshift variants. Which one is the missense? Please give the results of the prediction programs for the intronic/assumed splice site variants.
  10. Table 3: at least some of these papers are about molecularly conformed patients with CHARGE syndrome. Since the authors state in lines 250-251 differently, please indicate in the table how many of the patients were not molecularly confirmed.
  11. Line 256: refer to a more recent paper (see also point 14)
  12. Lines 262-264: is there indeed a different splice site prediction of variants affecting positions different form ±1/±2? Please give this information. Actually we do not see this effect in our patients, patients with far intronic variants being equally clinically affected compared to other splice site variants.
  13. Lines 300-304: this indeed is a nice information and is in agreement to what we see for other sensory impairments in CHARGE syndrome, e.g. individuals learn to compensate for absence of their vestibular balance. However, we need to take into account here the cognitive level of the patients as well. Did the authors look at the effect of cognitive impairment on visual compensation? It would be nice if they could add the intellectual development classification to table 2.
  14. Some recent publications on the phenotype (PMID: 20301533), the guidelines for clinical management PMID: 28160409 (,PMID: 29168326) and the observed CHD7 variants in CHARGE syndrome (PMID: 22461308) are missing.
  15. Some of the references seem to be incomplete, e.g. 20 and 22

Author Response

Thank you for revising the paper, we have carefully considered all issues and suggestions raised by you.

We provide a point-by-point response to your comments.

Round 2

Reviewer 1 Report

All my previous comments have been properly addressed.

Author Response

Thank you for your time

Reviewer 2 Report

I am happy with most of the changes made. Still some remarks remain:

Line 25: I do not know why the authors changed “molecularly confirmed” in “CHD7-mutation positive”. Mutation positive is confusing, because mutation is not specific enough. I suppose the authors mean a carrying a pathogenic CHD7 variant.

Line 384: mutations --> variants (and also elsewhere in the text)

Line 431: relative --> relatively

Table 2: I suppose the 2nd row should start with “Age (years)”
It is difficult to read the table with all changes. It seems that part of the CHD7 variants have been deleted. The c. notations disappeared for most variants. I recommend the authors to have a molecular geneticist carefully checking this row in table 2.

The English grammar still needs editing.

Author Response

Thank you for your observations.

We provide a point-by-point response to your comments.

Reviewer 2

  • Line 25: I do not know why the authors changed “molecularly confirmed” in “CHD7-mutation positive”. Mutation positive is confusing, because mutation is not specific enough. I suppose the authors mean a carrying a pathogenic CHD7 variant.

ANSWER. THIS HAS BEEN AMENDED

  • Line 384: mutations --> variants (and also elsewhere in the text)

ANSWER. THIS HAS BEEN AMENDED

  • Line 431: relative --> relatively

ANSWER. THIS HAS BEEN AMENDED

  • Table 2: I suppose the 2nd row should start with “Age (years)”
    It is difficult to read the table with all changes. It seems that part of the CHD7 variants have been deleted. The c. notations disappeared for most variants. I recommend the authors to have a molecular geneticist carefully checking this row in table 2.

ANSWER. THIS HAS BEEN AMENDED

  • The English grammar still needs editing.

ANSWER. THIS HAS BEEN PERFORMED      
